# Formulation and Biological Evaluation of Mesoporous Silica Nanoparticles Loaded with Combinations of Sortase A Inhibitors and Antimicrobial Peptides

**DOI:** 10.3390/pharmaceutics14050986

**Published:** 2022-05-04

**Authors:** Sitah Alharthi, Zyta M. Ziora, Taskeen Janjua, Amirali Popat, Peter M. Moyle

**Affiliations:** 1School of Pharmacy, The University of Queensland, Pharmacy Australia Centre of Excellence, Woolloongabba, QLD 4102, Australia; sitah.alharthi@uq.net.au (S.A.); t.janjua@uq.edu.au (T.J.); a.popat@uq.edu.au (A.P.); 2Department of Pharmaceutical Science, School of Pharmacy, Shaqra University, Riyadh 11961, Saudi Arabia; 3Institute for Molecular Bioscience (IMB), The University of Queensland, Saint. Lucia, QLD 4072, Australia; z.ziora@uq.edu.au; 4Mater Research Institute, The University of Queensland, Translational Research Institute, 37 Kent St., Woolloongabba, QLD 4102, Australia

**Keywords:** antimicrobial peptides, antimicrobial resistance, mesoporous silica nanoparticles, sortase A inhibitors, synergy

## Abstract

This study aimed to develop synergistic therapies to treat superbug infections through the encapsulation of sortase A inhibitors (SrtAIs; *trans*-chalcone (TC), curcumin (CUR), quercetin (QC), or berberine chloride (BR)) into MCM-41 mesoporous silica nanoparticles (MSNs) or a phosphonate-modified analogue (MCM-41-PO_3_^−^) to overcome their poor aqueous solubility. A resazurin-modified minimum inhibitory concentration (MIC) and checkerboard assays, to measure SrtAI synergy in combination with leading antimicrobial peptides (AMPs; pexiganan (PEX), indolicidin (INDO), and [I5, R8] mastoparan (MASTO)), were determined against methicillin-sensitive (MSSA) and methicillin-resistant (MRSA) *Staphylococcus aureus*, *Escherichia coli*, and *Pseudomonas aeruginosa*. The results demonstrated that the MCM-41 and MCM-41-PO_3_^−^ formulations significantly improved the aqueous solubility of each SrtAI. The MICs for SrtAI/MCM-41-PO_3_^−^ formulations were lower compared to the SrtAI/MCM-41 formulations against tested bacterial strains, except for the cases of BR/MCM-41 and QC/MCM-41 against *P. aeruginosa*. Furthermore, the following combinations demonstrated synergy: PEX with TC/MCM-41 (against all strains) or TC/MCM-41-PO_3_^−^ (against all strains except *P. aeruginosa*); PEX with BR/MCM-41 or BR/MCM-41-PO_3_^−^ (against MSSA and MRSA); INDO with QC/MCM-41 or QC/MCM-41-PO_3_^−^ (against MRSA); and MASTO with CUR/MCM-41 (against *E. coli*). These combinations also reduced each components’ toxicity against human embryonic kidney cells. In conclusion, MCM-41 MSNs provide a platform to enhance SrtAI solubility and demonstrated antimicrobial synergy with AMPs and reduced toxicity, providing novel superbug treatment opportunities.

## 1. Introduction

Antimicrobial-resistant pathogens represent a serious global threat to human health and economies. Owing to the global spread of resistant pathogens, current antimicrobial agents are losing their efficacy to treat or prevent important infections. Consequently, the morbidity and mortality associated with infectious diseases is increasing due to treatment failures [1]. The development of alternative therapies, which target virulence factors in antimicrobial-resistant bacteria, is an area of increasing interest. These so-called anti-virulence approaches block virulence-associated pathways, without affecting pathogen viability, and thus apply less pressure for the development of antimicrobial resistance [2]. Sortase A (SrtA) is a membrane-bound cysteine transpeptidase which is associated with Gram-positive bacteria. It plays a crucial role in attaching virulence-associated proteins to the bacterial cell wall. This is achieved through the recognition of an LPXTG (X = any amino acid) motif in the virulence-associated protein, followed by cleavage between the threonine and glycine and subsequent covalent attachment to a cell wall peptidoglycan-associated poly-glycine sequence [3]. SrtA is considered to be an ideal target for the development of anti-virulence agents because: (i) it is an extracellular membrane-anchored enzyme, and thus is readily accessible for drug targeting; (ii) it is not essential for bacterial viability; and (iii) it is absent in eukaryotic cells [4]. Thus, selective efficacy towards bacteria is possible [5].

Numerous naturally derived compounds have been identified as SrtA inhibitors (SrtAIs) [6], providing access to a wide range of potential novel antimicrobial therapies [7]. In this work, four of these naturally derived SrtAIs (*trans*-chalcone (TC), curcumin (CUR), quercetin (QC), and berberine chloride (BR)), which exhibit poor aqueous solubility that affects their utility as antivirulence agents, were selected for nanoencapsulation. Studies have demonstrated that the nanoencapsulation of similar hydrophobic compounds offers a promising approach to enhancing their solubility, permeability, and oral bioavailability, and consequently antibacterial activity [8]. Among literature-defined nanoencapsulation materials, mesoporous silica nanoparticles (MSNs) have been widely used for various biomedical applications due to their proven biocompatibility in human clinical trials [9,10], degradability, adjustable morphology, and ease of modification (e.g., the manipulation of particle size and surface charge [11]) to improve their drug loading characteristics [12].

MSNs with a uniform pore size and a long-range ordered mesoporous structure were first introduced by Mobil corporation scientists in 1992 as Mobil Composition of Matter Number 41 (MCM-41) [13], consisting of cylindrical mesopores with a typical 2 to 10 nm pore diameter [14]. MCM-41 offers many advantages for drug delivery, including: (i) a long-range ordered porous structure, without interconnection of individual pores, allowing for controlled drug loading [15]; (ii) a large pore volume (~1 cm^3^/g) and surface area (>1000 m^2^/g), providing high molecule loading and enhanced dissolution potentials [16]; and (iii) two functional surfaces (cylindrical pores and the exterior surface), which can be selectively functionalised to provide enhanced control over drug-loading and release, or, in the case of the external surface, can be conjugated to targeting ligands to provide efficient cell-specific drug delivery [16]. Previously, our research group and others have shown that different MSN surface functional groups can influence the activity of drugs loaded within the MSN nanopores and the stability of the formulation [17,18,19]. To date, there has been no extensive comparative investigation of the loading of each member of our SrtAI library into MSNs and how MSN loading affects their aqueous solubility, antimicrobial activity, and toxicity.

Herein, in an attempt to overcome the solubility issues associated with each of the four SrtAI library compounds (TC, CUR, QC, and BR) through nanoencapsulation, improve their antimicrobial activity, and reduce their toxicity compared to formulations with organic co-solvents [20], the loading of anionic MCM-41 and phosphonate-modified MCM-41 (MCM-41-PO_3_^−^) with each SrtAI was investigated, along with the effects of SrtAI loading on particle size, charge, morphology, and SrtAI aqueous solubility. It was hypothesized that these hydrophobic SrtAIs would be confined within the MCM-41 and MCM-41-PO_3_^−^ pores in an amorphous form, helping to improve their aqueous solubility [21]. In addition, in order to find combinations that improve the potency of AMPs, the antimicrobial activity of each SrtAI/MSN formulation was assessed for their capacity to synergistically enhance the antimicrobial activity of a library of leading antimicrobial peptides (AMPs; pexiganan (PEX), indolicidin (INDO), and [I5, R8] mastoparan (MASTO)) and reduce their toxicity towards mammalian cells. This is of significance as AMPs have a novel mechanism of action (in comparison to current antibiotics) which omits the development of resistance, but, due to their peptide nature, they exhibit significant costs. Thus, improving their potency would be of significance, as this would reduce the dose of an AMP required for efficacy, reducing their cost and toxicity, and, by encapsulating the AMP, would help protect them from degradation. Thus, these characteristics would improve the commercial viability of AMPs as alternative antimicrobial treatments for superbug infections.

## 2. Materials and Methods

### 2.1. Materials

The bacterial strains (methicillin-sensitive *S. aureus* (MSSA) ATCC 25923, methicillin-resistant *Staphylococcus aureus* (MRSA) ATCC43300, *E. coli* ATCC 25922, and *P. aeruginosa* ATCC 27853) were obtained from the American Type Culture Collection (Manassas, VA, USA). All amino acids and Rink amide-4-methylbenzhydrylamine (MBHA) resin (0.54 mmol/g) were obtained from Mimotopes (Melbourne, VIC, Australia). Ethyl (hydroxyimino)cyanoacetate (Oxyma Pure) and *N*,*N′*-diisopropylcarbodiimide (DIC) were obtained from Chem-Impex International (Wood Dale, IL, USA). Triisopropylsilane (TIPS) was obtained from Alfa Aesar (Massachusetts, VA, USA). Dulbecco’s Modified Eagle Medium (DMEM), fetal bovine serum (FBS), and penicillin/streptomycin were obtained from Thermo Fisher Scientific (Scoresby, VIC, Australia). Acetonitrile (MeCN), cetyltrimethylammonium bromide (CTAB, 99%), *N,N*-dimethylformamide (DMF; peptide synthesis grade), dimethyl sulfoxide (DMSO), ethanol (EtOH), methanol (MeOH), Mueller–Hinton broth (MHB), tetraethyl orthosilicate (TEOS, reagent grade 98%), trifluoroacetic acid (TFA), 3-(trihydroxysilyl)propyl methylphosphonate (THMP, 90%), and all other reagents were purchased from Merck (Castle Hill, NSW, Australia).

### 2.2. Equipment

Deionized reverse osmosis ultrapure water (18 MΩ) was prepared by a Millipore Simplicity UV ultrapure water system. A Biotage^®^ Initiator^+^ Alstra™ (Biotage, Uppsala, Sweden) microwave peptide synthesizer was used for microwave-assisted peptide synthesis. LC-MS was performed on a Shimadzu LCMS-2020 using LabSolutions 5.89 software (Shimadzu, Kyoto, Japan). Analytical reversed-phase high-performance liquid chromatography (RP-HPLC) was performed on an Agilent 1200 series analytical HPLC (G1379B degasser, G1312A binary pump, G1329A ALS autosampler, G1316A thermostatted column compartment, G1315D diode array detector, and Agilent Chemstation Rev.B.04.02 software; Agilent, Santa Clara, CA, USA). Preparative RP-HPLC was conducted on an Agilent 1200 Series Preparative Scale HPLC (G1361A preparative pumps, G1364B fraction collector, G1365D multi-wavelength detector, 3725i-038 Rheodyne preparative scale manual sample injector, and Agilent Chemstation Rev.B.04.02 software). Solvent A (0.1% (*v*/*v*) TFA-water) and solvent B (90% (*v*/*v*) MeCN/0.1% (*v*/*v*) TFA-water) were used as mobile phases in linear gradient mode over 30 min and 60 min for analytical and preparative separations, respectively. Separations were performed on Vydac C18 analytical (218TP54; 150 × 4.6 mm; 5 μm) or C18 preparative (218TP1022; 250 × 22 mm; 10 μm) columns at 1 mL/min or 10 mL/min respectively, with detection at 214 nm. Fluorescence readings were obtained on an Envision multilabel plate reader (PerkinElmer, Rowville, VIC, Australia). A Thermolyne muffle furnace (Thermo Scientific, Asheville, NC, USA) was used for calcination of MCM-41. A Heraeus Multifuge X1R^®^ Centrifuge (Thermo Scientific, Langenselbold, Germany) was used to pellet and collect MCM-41 and MCM-41-PO_3_^−^ MSNs. A Buchi Rotavapor^®^ R-100 rotary evaporator (Buchi, Flawil, Switzerland) was used for loading MCM-41 and MCM-41-PO_3_^−^ MSNs with SrtAIs. A Malvern Zetasizer Nano ZS (Malvern Panalytical, Malvern, UK) was used to measure the intensity-weighted mean hydrodynamic diameter (Z-Ave), polydispersity index (PDI), intensity mean, and Zeta-potential. A Mettler Toledo^®^ (Columbus, OH, USA) thermogravimetric analysis (TGA)/differential scanning calorimeter (DSC) was used to investigate the loading capacity of MCM-41 and MCM-41-PO_3_^−^ MSNs. Transmission electron microscopy (TEM) was performed on a Hitachi HT7700B at 100 kV to analyse the shape and size of the particles. Micromeritics Tri-StarII^®^ (Norcross, GA, USA) was used for N_2_-physisorption isothermal analysis of the surface area and porosity. A Perkin Elmer Spectrum TWO (Liantrisant, UK) Fourier-transform infrared spectrometer (FT-IR) was used to investigate interactions between MSNs and SrtAIs.

### 2.3. Preparation of MCM-41 MSNs

MCM-41 nanoparticles were prepared according to a previous report [21], with slight modifications (Figure 1). Briefly, 1 g of CTAB was dissolved in 480 mL of 18 MΩ water and vigorously stirred (700 RPM) at room temperature (RT) to obtain a clear solution. Subsequently, 3.5 mL of 2 M NaOH was added, and the mixture was heated to 80 °C with a silicon oil bath, followed by the dropwise addition of TEOS (6.7 mL) to the mixture and stirring (700 RPM) for 2 h. The MCM-41 particles were harvested by vacuum filtration using filter paper (Whatman™, 150 mm) and washed with 200 mL of 18 MΩ water and then 200 mL of EtOH. The particles were dried (60 °C) overnight, and calcination was then conducted in a muffle furnace (2 h to reach 550 °C, 5 h at 550 °C; the temperature slowly decreased overnight to reach RT).

### 2.4. Functionalization of MCM-41 with PO_3_^−^

MCM-41 nanoparticles were modified with a phosphonate functional group (PO_3_^−^) (Figure 1) based on a reported procedure [19]. Briefly, 400 mg of MCM-41 was suspended in 18 MΩ reverse osmosis water (65 mL) plus THMP (400 μL) at pH 5 (pH measured with pH test strips and adjusted with 1 M HCl) and refluxed at 100 °C overnight. The particles were then collected by centrifugation (16,000× *g*, 15 min, RT), and the obtained pellet was washed twice with 10 mL of 18 MΩ water and then twice with 10 mL of EtOH. MCM-41-PO_3_^−^ MSNs were then dried in an oven at 60 °C overnight and stored in a vacuum desiccator at RT.

### 2.5. Loading MCM-41 and MCM-41-PO_3_^−^ MSNs with SrtAIs

SrtAIs (TC, CUR, BR, and QC) were individually loaded into MCM-41 and MCM-41-PO_3_^−^ (Figure 1) by rotary evaporation, as previously described [22], with modifications. For this purpose, individual SrtAIs (10 mg) were dissolved in MeOH (8 mL) using sonication (60 W; Branson CPX2800H-E; Branson Ultrasonics, Brookfield, CT, USA), into which MCM-41 or MCM-41-PO_3_^−^ (90 mg) were dispersed, sonicated (60 W; 5 min), and stirred (400 RPM) overnight at RT. MeOH was then evaporated (40 °C) using a rotary evaporator, and the obtained powders were dried at RT and stored in the dark at 4 °C.

### 2.6. Characterization of SrtAI-Loaded MCM-41 and MCM-41-PO_3_^−^

MCM-41 and MCM-41-PO_3_^−^ morphology was determined by TEM at an accelerating voltage of 80 kV. For this, SrtAI-loaded and non-loaded MCM-41 and MCM-41-PO_3_^−^ (1 mg) were individually dispersed in EtOH (1 mL) and sonicated (2 min). One drop was then cast on a carbon-coated copper grid and allowed to air dry before imaging. DLS was also performed to determine SrtAI-loaded and non-loaded MCM-41 and MCM-41-PO_3_^−^ Z-Ave, PDI, intensity mean, and Zeta-potential. For this purpose, samples (1 mg) were dispersed in PBS (1 mL; pH 7.4), bath sonicated (5 min), and measured in disposable folded capillary zeta cells (DTS1070, Malvern Panalytical, Malvern, UK). TGA and DSC measurements were performed to investigate the thermal degradation of each component and to determine the loading capacity. For each SrtAI, 5 mg was deposited into a TGA 70 µL alumina crucible. A temperature range of 50 (25 for TC) to 900 °C, a heating rate of 10 °C min^−1^, and an airflow of 20 mL/min were used. N_2_-Physisorption isothermal analysis was performed to determine MSN pore volume, size, and surface area. The Brunauer–Emmeyt–Teller (BET) theory was used to determine surface area. The pore size distribution was measured using the BJH model from the isotherm adsorption branch. Prior to analysis, SrtAI-loaded and unloaded MCM-41 and MCM-41-PO_3_^−^ (70 mg) were degassed for 10 h at 50 °C. FT-IR spectra were obtained for 2 mg of each sample (SrtAIs, MSNs, and SrtAI-loaded MSNs) over a 4000 to 400 cm^−1^ wavenumber range to investigate interactions between MSNs and SrtAIs (data provided in Appendix A).

The solubility of unformulated SrtAIs and MCM-41 and MCM-41-PO_3_^−^ formulated SrtAIs were assessed using an established protocol [21], with modifications. Briefly, saturated solutions of SrtAIs and SrtAI-loaded MCM-41 and MCM-41-PO_3_^−^ were prepared by dispersing SrtAIs and SrtAI-loaded MCM-41 and MCM-41-PO_3_^−^ individually in 18 MΩ water at a concentration of 2 mg/mL, followed by stirring for 24 h in the dark at 37 °C. The solutions were then centrifuged for 10 min at 150 RPM at RT. The resulting supernatant was collected, and the area under the curve (AUC) was determined at 280 nm for TC, CUR QC, and BR using RP-HPLC. The concentrations were then calculated based on a calibration curve. All measurements were performed in triplicate, and the mean value was calculated.

### 2.7. Synthesis of Antimicrobial Peptides

Each AMP was synthesized by fluorenylmethoxycarbonyl (Fmoc)-solid-phase peptide synthesis (SPPS) on Rink amide-MBHA resin (0.56 mmol/g; 0.125 mmol). For coupling amino acids, an automated microwave peptide synthesizer (Biotage^®^ Initiator^+^ Alstra™) was used. DIC/Oxyma Pure couplings were performed using DIC (4 eq.), Oxyma Pure (4 eq.), and Fmoc-amino acids (4 eq.) in DMF. Fmoc-deprotection steps were performed by incubating the resin in 20% (*v*/*v*) piperidine-DMF for 3 min at RT, and then repeated for 10 min. After swelling the resin in DMF (20 min, 70 °C), an Fmoc-deprotection step was performed. Amino acids were then double coupled (75 °C, 5 min for all amino acids except arginine, which was coupled at RT, 60 min) to assemble the AMP sequences (PEX: GIGKFLKKAKKFGKAFVKILKK-NH_2_; INDO: ILPWKWPWWPWRR-NH_2_; MASTO: INLKILARLAKKIL-NH_2_).

PEX, INDO, and MASTO were cleaved from each resin using 95% (*v*/*v*) TFA-2.5% (*v*/*v*) TIPS-2.5% (*v*/*v*) water (10 mL/g resin) for 2 h at RT. Subsequently, the mixture volumes were reduced using a nitrogen gas stream, and ice-cold diethyl ether was added to precipitate the peptides, followed by pelleting by centrifugation. After removing the supernatant, the precipitated peptides were dissolved in 50% (*v*/*v*) MeCN–0.1% (*v*/*v*) TFA-water. The mixture was filtered and freeze-dried to harvest the crude peptides.

Preparative HPLC was used to purify the crude peptides, and pure fractions were combined and freeze-dried to obtain purified peptides. To evaluate the purity and identity of the peptides, RP- HPLC and mass spectrometry were used. For each peptide, two RP-HPLC gradients were applied: 0–70% solvent B as a wide gradient for all peptides and 10–50, 25–45, and 30–50% solvent B as narrow gradients for PEX, INDO, and MASTO, respectively.

### 2.8. Preparation of Bacterial Inoculums

Bacterial inoculums were prepared according to the Clinical and Laboratory Standards Institute (CLSI) [23]. From overnight *S. aureus* (MSSA) ATCC25923, MRSA ATCC43300, *E. coli* ATCC25922, and *P. aeruginosa* ATCC27853 *agar* cultures, two colonies were suspended in individual vials containing sterile normal saline (2 mL). The turbidity of each bacterial suspension stock was then adjusted to equivalence with the 0.5 McFarland standard (1 × 10^8^ colony-forming units/mL; CFU/mL) using sterile normal saline, then further diluted to 1 × 10^6^ CFU/mL for the broth microdilution (BMD) minimum inhibitory concentration (MIC) assay [23].

### 2.9. Antibacterial Activity of SrtAI-Loaded and Non-Loaded MCM-41 and MCM-41-PO_3_^−^

The MICs of each SrtAI-loaded MCM-41 and MCM-41-PO_3_^−^ formulation against four bacterial strains (*S. aureus* (MSSA) ATCC25923, MRSA ATCC43300, *E. coli* ATCC25922, and *P. aeruginosa* ATCC27853) were determined using the CLSI BMD method [24] with a resazurin colorimetric readout [20]. For this method, quantities of SrtAI-loaded MCM-41 or MCM-41-PO_3_^−^ containing 1 mg of individual SrtAIs were suspended in MHB (to 1 mL) to yield a 1 mg/mL final SrtAI concentration. From these stocks, 200 µL/well was dispensed into column one of a sterile 96-well clear, round-bottomed polystyrene microplate. In columns two-to-eleven, 100 µL/well of MHB was dispensed, and in column twelve, 200 µL/well of MHB was dispensed. Subsequently, a 2-fold serial dilution of the SrtAI-loaded MCM-41 or MCM-41-PO_3_^−^ MSNs was prepared by transferring 100 µL from the first column, mixing this with the next column to the right, and repeating this process to column ten, with 100 mL of the mixture from column ten then discarded. Subsequently, 100 µL of the diluted standardized inoculum (1 × 10^6^ CFU/mL) was added into the wells in columns one-to-eleven to obtain the final volume of 200 µL (5 × 10^5^ CFU/mL) and final SrtAI concentrations ranging from 500 to 1 µg/mL. Column eleven (+) contained no SrtAI formulation and served as a control for bacterial growth. Column twelve (−) contained only MHB as a sterility control. The plates were then incubated at 37 °C for 24 h, and the wells visibly investigated for turbidity. Resazurin solution (30 µL; 0.15 mg/mL in sterile water) [25] was added to each well, and the plates were incubated for 2 h at 37 °C. The MIC values were visually determined as the lowest concentration that retained the dark blue resazurin colour [25].

### 2.10. Antibacterial Activity of AMPs

The MICs of PEX, INDO, and MASTO against *S. aureus* ATCC25923, MRSA ATCC43300, *E. coli* ATCC25922, and *P. aeruginosa* ATCC27853 were determined according to the CLSI BMD method with resazurin readout [20], as described above. In this method, 100, 200, and 300 µg of PEX, INDO, MASTO, respectively, were dissolved in 0.01% (*v*/*v*) DMSO-MHB (to 1 mL) and used to load column one of individual 96-well microplates. After 2-fold serial dilution and the addition of the standardized inoculum (100 µL; 1 × 10^6^ CFU/mL), PEX, INDO, and MASTO yielded 50.0–0.10 µg/mL, 150–0.29 µg/mL, and 100–0.19 µg/mL concentration ranges, respectively. Experiments were performed in triplicate.

### 2.11. Checkerboard Test

To investigate the potential for synergistic, additive, or antagonistic effects when combining SrtAI-loaded MCM-41 or MCM-41-PO_3_^−^ with AMPs, a checkerboard test was performed to determine the fractional inhibitory concentration index (ΣFIC) according to a previously reported protocol [26], with modifications. For this purpose, SrtAI-loaded MCM-41 or MCM-41-PO_3_^−^ equivalent to 2 mg of each SrtAI were suspended in MHB (to 1 mL), and PEX (200 µg), INDO (600 µg), and MASTO (400 µg) were individually mixed with 0.01% (*v*/*v*) DMSO-MHB (to 1 mL) to produce antimicrobial stocks. From each stock, six additional concentrations were prepared using a 2-fold serial dilution. To prepare the checkerboard assay, 50 µL/well of the highest concentration of each AMP was transferred into column one of individual sterile 96-well round-bottomed polystyrene plates, followed by their 2-fold dilution series in columns two-to-seven. In column eight, 50 µL/well of 0.01% (*v*/*v*) DMSO-MHB was added. Subsequently, 50 µL/well of the highest concentration SrtAI-loaded MCM-41 or MCM-41-PO_3_^−^ MSNs was added to row A, followed by their 2-fold dilution series in rows B-to-G. In row H, 50 µL/well of MHB was added. Subsequently, 100 µL of the bacterial inoculum (1 × 10^6^ CFU/mL) was added to each well to obtain a 5 × 10^5^ CFU/mL final concentration. The plates were then incubated at 37 °C for 24 h, followed by the addition of resazurin solution (30 µL; 0.15 mg/mL in sterile water) to each well. The plates were incubated for 2 h at 37 °C. MIC values were visually determined as the lowest concentration that retained the dark blue resazurin colour in column eight for SrtAIs and in row H for AMPs. To determine if combinations provide additive, synergistic, or antagonistic effects, FIC index (ΣFIC) values were calculated using the following formula:(1)$$\sum \;\mathrm{FIC}=\frac{\mathrm{MIC}\;\mathrm{of}\;\mathrm{AMP}\;\mathrm{in}\;\mathrm{combination}}{\mathrm{MIC}\;\mathrm{of}\;\mathrm{AMP}\;\mathrm{alone}}\;+\;\frac{\mathrm{MIC}\;\mathrm{of}\;\mathrm{SrtAI}\;\mathrm{loaded}\;\mathrm{MSNs}\;\mathrm{in}\;\mathrm{combination}}{\mathrm{MIC}\;\mathrm{of}\;\mathrm{SrtAI}\;\mathrm{alone}}$$

The combination used to calculate the ΣFIC value represents the well/s where the largest change from the MIC value of one of the antimicrobial agents was observed. Combinations are defined as synergistic where ΣFIC is ≤ 0.5, additive where ΣFIC is > 0.5 to 1, indifferent where ΣFIC is > 1 to 4, and antagonistic where ΣFIC is > 4 [27].

### 2.12. Cell Viability

The effects of each SrtAI-loaded MCM-41 or MCM-41-PO_3_^−^ MSN, AMP, and their combinations on HEK-293 cell viability were assessed using a resazurin reduction assay [28]. The concentrations of each antimicrobial used in these assays was selected based on their MIC and ΣFIC data from the BMD and checkerboard assays against MSSA, respectively. For this purpose, HEK-293 cells (5000 cells/well) were seeded in DMEM media supplemented with 10% (*v*/*v*) FBS and 100 U/mL penicillin G-100 µg/mL streptomycin (complete media) in 96 well flat-bottomed, black polystyrene plates. The cells were then incubated (37 °C, 5% CO_2_) for 48 h to reach 80% confluency. The media was then discarded, and the cells were individually treated with each SrtAI-loaded MSN, AMPs, or their combinations, at different concentrations in 100 µL of complete media containing 0.01% (*v*/*v*) DMSO. Cells treated with complete media containing 0.01% (*v*/*v*) DMSO or with 10% (*v*/*v*) SDS and 0.1 M HCl were used as negative and positive toxicity controls, respectively. Wells containing complete media with 0.01% (*v*/*v*) DMSO were used as a background control. The plates were then incubated (37 °C, 5% CO_2_) for 24 h, the supernatant was removed, 50 μL of 100 μM resazurin in PBS was added into each well, and the plates were incubated (37 °C, 5% CO_2_) for 4 h. Resorufin fluorescence (Fl) was read at 590 nm, with excitation at 560 nm [28]. The percentage of the viable cells was calculated according to the following formula:(2)$$\%\;\mathrm{Cell}\;\mathrm{viability}=\frac{\mathrm{Fl}\;\mathrm{sample}\;-\;\mathrm{Fl}\;\mathrm{background}}{\mathrm{Fl}\;\mathrm{negative}\;\mathrm{control}\;-\;\mathrm{Fl}\;\mathrm{background}\;}\times 100.$$

Experiments were performed in triplicate to demonstrate reproducibility.

### 2.13. Statistical Analysis

For statistical analyses, GraphPad Prism software version 9.3 was used. Cell viability data and solubility analysis are presented as the mean ± standard deviation (SD, *n* = 3). One-way analysis of variance (ANOVA) followed by Tukey’s post-hoc test was used to analyse for statistical differences.

## 3. Results and Discussion

### 3.1. Characterization of SrtAI-loaded MCM-41 and MCM-41-PO_3_^−^ MSNs

DLS was used to measure the particle size (Z-Ave, intensity mean), polydispersity (PDI), and surface charge (Zeta-potential) of the MCM-41 and MCM-41-PO_3_^−^ MSNs before and after loading with SrtAIs, with the results presented in Table 1. No significant difference with respect to Z-Ave (120.8 ± 2.30 and 128.8 ± 3.20 nm, respectively) and intensity mean (143.9 ± 2.60 and 138.9 ± 3.80, respectively) were observed between the unloaded MCM-41 and MCM-41-PO_3_^−^ MSNs. However, the MCM-41-PO_3_^−^ MSNs exhibited a lower PDI (0.07 ± 0.01) and a more negative Zeta-potential (−49.0 ± 1.90 mV) compared to the MCM-41 MSNs (0.16 ± 0.03 and −41.1 ± 0.90 mV, respectively).

The loading of SrtAIs into the MSNs affected the Z-Ave size and PDI. For MCM-41, a slightly lower Z-Ave size was observed for TC (107.8 ± 1.70 nm), CUR (105.9 ± 2.00 nm), and QC (105.9 ± 2.00 nm)-loaded MSNs, while a higher Z-Ave was observed for BR (141.8 ± 1.80 nm)-loaded MSNs (Table 1 and Figure 1). In comparison, similar Z-Ave sizing (125.4–127.4 nm) to the unloaded MSNs was observed for TC, CUR, and QC-loaded MCM-41-PO_3_^−^, with the BR-loaded MCM-41-PO_3_^−^ again demonstrating a larger Z-Ave size (162.5 ± 1.80 nm) (Table 1 and Figure 1). The polydispersity index (PDI) was also observed to increase with SrtAI-loading, with the MCM-41-PO_3_^−^ MSNs continuing to demonstrate lower PDI values (0.16–0.22) compared to MCM-41 (0.19–0.29) after SrtAI loading (Table 1). Despite this, the observed PDI values for the SrtAI-loaded MSNs were similar (0.16–0.22), except for CUR-loaded MCM-41 (0.29 ± 0.01), which exhibited a higher variation in particle size. However, all formulations remained dispersed without large aggregates (PDI < 0.3) [29]. Finally, SrtAI-loading was observed to reduce the negative Zeta-potential associated with each MSN, with TC and BR having the greatest effect. For BR, this was hypothesized to be due in part to ionic interactions between its positively charged quaternary ammonium group and the negatively charged MSN silanol and phosphonate groups.

The morphology of each MSN nanoformulation was also visualised by TEM before and after (Figure 2) SrtAI loading. The nanoparticles were approximately spherical in shape. The morphology was similar between the MCM-41 and MCM-41-PO_3_^−^ MSNs, suggesting that phosphonate modification did not affect the particle shape. The MSN size was also evaluated by TEM (Figure 2), with particle sizes appearing similar in magnitude to the DLS sizing data (Table 1).

TGA was used to determine the SrtAI loading capacity of MCM-41 and MCM-41-PO_3_^−^ MSNs by comparing the weight loss associated with temperature ramping (50–900 °C; 25–900 °C for TC-loaded MSNs) between unloaded MSNs, SrtAI-loaded MSNs, and the SrtAIs alone. Using this technique (Figure 3), the loading capacities of TC, CUR, QC, and BR in MCM-41 MSNs were 9.8, 7.6, 9.6, and 8.7% (*w*/*w*), respectively, and 9.2, 9.3, 8.5, and 6.2% (*w*/*w*) in MCM-41-PO_3_^−^ MSNs, respectively. The theoretical MSN loading capacity for each SrtAI was 10% (*w*/*w*), based on the rotary evaporation formulation method, where 10 mg of each individual SrtAI was added to 90 mg of each MSN. Thus, these data suggest that the MSNs were efficiently loaded with SrtAIs. In addition, similar SrtAI loading capacities were measured between the MCM-41 and MCM-41-PO_3_^−^ MSNs, except for BR, which demonstrated a 29% lower loading capacity with MCM-41-PO_3_^−^ compared to MCM-41. In agreement with this study, other studies [19,30] have demonstrated the capacity of the rotary evaporation loading method to effectively encapsulate both hydrophilic and hydrophobic drugs within the extensive porous network of MSNs.

To further characterise the crystallinity of each SrtAI when loaded within MSNs, DSC was performed [31]. The investigation of each individual SrtAI by DSC revealed endothermic peaks at 61 (TC), 177 (CUR), 317 (QC), and 190 °C (BR) (Figure 3), which correspond to reported melting points for the crystalline form of each SrtAI [32]. Physical mixtures of non-loaded MSNs with each individual SrtAI, with the exception of QC, demonstrated these same peaks. In comparison, after loading the SrtAIs into MSNs, these peaks were greatly reduced or disappeared (Figure 3), suggesting that the SrtAIs were incorporated into the nanoparticles in an amorphous form [33].

To characterize the surface area, pore size, and volume of prepared MSNs, N_2_-adsorption-desorption isotherms were acquired for SrtAI-loaded and non-loaded MSNs. The results (Table 2 and Appendix A) were characteristic of an IV isotherm with a steep capillary condensation at a relative pressure (P/Po) range of 0.2–0.4, in loaded and unloaded particles, which is a characteristic of MCM-41 [34,35]. The surface area for MCM-41 was calculated as 892.6 m^2^/g, with a calculated pore volume of 0.9 cm^3^/g and pore size of 1.9 nm. After phosphonate modification, a decrease in the surface area (381.6 m^2^/g) and pore volume (0.6 cm^3^/g) was observed, with no effect on pore size (1.9 nm) for unloaded particles. After loading with SrtAIs, the surface area was also observed to decrease to 459.7, 421.6, 500.6, and 559.3 m^2^/g for TC, CUR, QC, and BR-loaded MCM-41, respectively, while the effects on the surface area of MCM-41-PO_3_^−^ were more variable, with TC and QC-loaded particles showing surface area reductions (215.2 and 215.2 m^2^/g, respectively). In contrast, CUR and BR-loaded MCM-41-PO_3_^−^ showed no effect or an increase in surface area (399.6 and 439.7 m^2^/g), respectively. Finally, the pore volume for TC, CUR, QC, and BR-loaded MCM-41 and MCM-41-PO_3_^−^ was reduced (Table 2). This reduction in pore volume, in combination with the reductions in surface area observed after SrtAI-loading for most formulations and TGA/DSC data (Figure 3) demonstrating the presence of each SrtAI and a loss of crystallinity for most formulations, means that it is highly likely that the SrtAIs were loaded within the mesopores.

SrtAI solubility was assessed in water to evaluate if SrtAI-loaded MSN formulations improved the dissolution of SrtAIs in water. The amount of each SrtAI that dissolved in water was significantly increased for each SrtAI MSN formulation (Figure 4), except for the QC-loaded MCM-41 formulation. The largest improvements in solubility were observed for the CUR (8.5-fold compared to CUR alone) and TC-loaded (34.7-fold compared to TC alone) MSNs, followed by BR-loaded MSNs (3.3-fold compared to BR alone). In comparison, minimal improvement was observed for the QC-loaded MSN formulations. Furthermore, no statistically significant differences in solubility were observed in the case of each SrtAI between their MCM-41 and MCM-41-PO_3_^−^ formulations. The enhanced solubility profile for each SrtAI-loaded MSN formulation is likely explained by the conversion of the compounds from a crystalline to an amorphous state upon MSN encapsulation, which has previously been demonstrated to improve the aqueous solubility of hydrophobic crystalline compounds [36].

### 3.2. Synthesis of Antimicrobial Peptides (AMPs)

The AMPs (PEX, INDO, and MASTO; Appendix A) were successfully synthesized in high yield (26, 25, and 33%, respectively) and purity (≥98% by HPLC area under the curve) by microwave-assisted Fmoc-SPPS on Rink amide-MBHA resin with DIC/Oxyma pure couplings and purified by preparative RP-HPLC.

### 3.3. Antibacterial Activity of SrtAI-Loaded MCM-41 or MCM-41-PO_3_^−^ and AMPs

A BMD assay, with a resazurin colorimetric readout [20], was performed to determine the MIC of each individual SrtAI-loaded (Figure 5) and non-loaded MSN and AMP (PEX, INDO, and MASTO) (Appendix A) against *S. aureus* (MSSA) ATCC25923, MRSA ATCC43300, *E. coli* ATCC25922, and *P. aeruginosa* ATCC27853 (Table 3). The results (Figure 5, Table 3) demonstrated that the MICs of SrtAI loaded MCM-41-PO_3_^−^ MSNs were in most cases 2-fold lower than the SrtAI-loaded MCM-41 MSNs; the AMPs demonstrated potent antimicrobial activity against each tested bacterial strain, including MRSA, in the general order MASTO > PEX > INDO; and the unloaded MSNs did not demonstrate antimicrobial activity in the tested concentration range (≤500 mg/mL). In addition, the CUR and QC-loaded MSN formulations were demonstrated to be more potent than their previously reported 5% (*v*/*v*) DMSO Muller–Hinton Broth formulations [20], ranging from 4- to 16-fold improvements in their MIC values, with the MCM-41-PO_3_^−^ MSN formulations in general being more potent. In comparison, a 2-fold maximum improvement was observed for TC and BR MCM-41-PO_3_^−^ MSNs, with no improvement seen for TC and BR MCM-41 formulations. Overall, these data suggested that the MSN formulations had the potential to improve the potency of some members of the SrtAI library, in particular CUR and QC, with the phosphonate modification also providing a small benefit in terms of MIC.

### 3.4. Checkerboard Assay

A checkerboard assay was performed to investigate the possibility of antimicrobial synergy between SrtAI-loaded MSNs and AMPs (PEX, INDO, and MASTO) against *S. aureus* (MSSA) ATCC25923, MRSA ATCC43300, *E. coli* ATCC25922, and *P. aeruginosa* ATCC27853. Such synergy would decrease the amount of AMP and/or SrtAI required for antimicrobial efficacy, potentially reducing the cost, adverse effects, and opportunities for antimicrobial resistance to develop. The results of this assay (Table 4 and Appendix A, with *S. aureus* ATCC25923 examples presented in Figure 6) are presented after the calculation of ΣFIC values from the well/s that demonstrated the largest change in the MIC values of one of the antimicrobial agents (FIC wells are indicated by orange and green boxes in Figure 6, with individual component MIC values indicated by yellow boxes). ΣFIC values ≤ 0.5 are representative of synergistic interactions, whereas values greater than 0.5 to 1 are additive and values greater than 1 to 4 are indifferent. From these results, the best synergy data was obtained where PEX was combined with SrtAI-loaded MSNs. Of these, the most synergistic combinations were observed for PEX with TC-loaded MCM-41 MSNs, which demonstrated synergy against all tested bacterial strains, with the MCM-41-PO_3_^−^ formulation displaying similar activity, although with additive rather than synergistic activity against *P. aeruginosa* (Table 4 and Appendix A). Similarly, BR-loaded MSNs demonstrated synergy against MSSA (Figure 6 and Table 4) and MRSA, with additive activity towards *E. coli*; however, this combination was borderline indifferent against *P. aeruginosa* (Table 4 and Appendix A). The combination of CUR and QC-loaded MSNs with PEX demonstrated further reductions in benefit (Table 4), with the CUR-loaded MSNs demonstrating additive effects against MSSA, MRSA, *E. coli*, and *P. aeruginosa* when formulated with MCM-41 (Appendix A), with indifferent effects against *P. aeruginosa* when the MCM-41-PO_3_^−^ formulation was used (Appendix A). In contrast, the QC-loaded MSNs demonstrated little benefit when combined with PEX (Table 4), with additive effects seen against MSSA (Figure 6), MRSA, and *E. coli* for the MCM-41-PO_3_^−^ formulation (Appendix A), but reduced to indifferent effects against *E. coli* for the MCM-41 formulation (Appendix A), and indifferent effects against *P. aeruginosa* for both MSN formulations (Appendix A) were observed. Synergy was also observed for the combination of MASTO with CUR-loaded MCM-41 MSNs against *E. coli* (Appendix A), with additive effects against all other assessed bacterial strains (Table 4 and Appendix A) and in the cases of INDO with QC-loaded MCM-41 and MCM-41-PO_3_^−^ MSNs against MRSA (Table 4 and Appendix A). In this second case, additive effects were observed against MSSA (Figure 6), with indifferent effects against *E. coli* and *P. aeruginosa* (Table 4 and Appendix A). Overall, many of the assessed combinations were synergistic or additive against the assessed bacterial library (Table 4), with the combination of BR-loaded MCM-41 MSNs with PEX providing the greatest synergy against the assessed bacterial library. This combination (Figure 6) reduced the MIC of BR towards MSSA by 16- (yellow box; 7.81 mg/mL) or 8-fold (orange box; 15.6 mg/mL) and PEX by 4- (yellow box; 1.57 mg/mL) or 8-fold (orange box; 0.79 mg/mL), depending on which synergistic combination was selected.

### 3.5. Cell Viability

The effects of SrtAI-loaded and non-loaded MSNs, AMPs (PEX, INDO, and MASTO), and their combinations on HEK-293 cell viability were evaluated using a resazurin-based assay [37] to determine if their synergistic combinations could reduce toxicity associated with individual components and if the phosphonate modification of MCM-41 MSNs affects toxicity. Concentrations (Appendix A) based on a two-fold dilution series, which ensure two concentrations below and three concentrations above the MIC (Table 3) or synergistic concentrations determined by ΣFIC (Table 4) against MSSA, were used.

The results of these experiments (Figure 7, Figure 8 and Appendix A) indicated a lack of toxicity for each component (Figure 7) and their combinations (Figure 8 and Appendix A) at their MIC or synergistic concentrations for MSSA. With respect to the individual AMPs, PEX and INDO did not affect cell viability at concentrations 8-fold (50 and 150 µg/mL, respectively) higher than their MICs for MSSA, while MASTO did not affect cell viability at a 4-fold (12.5 µg/mL) higher concentration than its MIC for MSSA. However, MASTO demonstrated approximately 20% toxicity at an 8-fold higher concentration (25 µg/mL) than this MIC value.

With respect to the SrtAI-loaded MSNs, CUR, QC, and BR-loaded MSNs did not affect cell viability (~100%; Figure 7) at their MICs for MSSA. However, TC-loaded MCM-41 MSNs demonstrated an approximately 20% decrease in HEK-293 cell viability at its MIC against MSSA (125 µg/mL). Of the different SrtAI-loaded MSN formulations, the TC-loaded MSNs demonstrated the greatest toxicity, with the MCM-41 and MCM-41-PO_3_^−^ MSN formulations demonstrating a >90% reduction in cell viability at 250 µg/mL (2- and 4-fold higher than their MIC values against MSSA, respectively). The BR-loaded MCM-41-PO_3_^−^ and MCM-41 MSNs demonstrated an approximately 50% reduction in cell viability at 250 and 500 µg/mL (4-fold higher than their MIC values against MSSA), respectively. In contrast, the CUR and QC-loaded MSNs demonstrated minimal effects on cell viability across the tested concentration ranges. This data suggests that CUR and QC-loaded MSNs may provide a less toxic means of inhibiting SrtA, with combinations of BR and TC-loaded MSNs that provide synergy providing a means to reduce these toxicities by allowing for delivery of lower SrtAI doses.

Finally, combinations of SrtAI-loaded MSNs with AMPs, which demonstrated synergy in the checkerboard assay, were assessed for their effects on HEK-293 cell viability. Based on this data (Figure 8), all the assessed combinations demonstrated no effects on cell viability at concentrations corresponding to the checkerboard assay well that was used to calculate the ΣFIC value. From this data, the combinations of TC or BR-loaded MCM-41-PO_3_^−^ MSNs with PEX or QC-loaded MCM-41-PO_3_^−^ MSNs with INDO demonstrated the smallest effects on cell viability (Figure 8B), with no significant effects observed at 4-fold above the FIC concentrations for the first two formulations and 8-fold above for the last. In contrast, the synergistic combinations of SrtAI-loaded MCM-41 MSNs with AMPs demonstrated a greater effect on cell viability (Figure 8A) than the MCM-41-PO_3_^−^ analogues, with the QC-loaded MCM-41 MSNs with INDO similarly demonstrating the least effect on cell viability, with no significant effect observed at 4-fold above the FIC concentrations. In contrast, a greater than 90% reduction in cell viability was observed at 8-fold above the FIC concentrations for TC-loaded MCM-41 MSNs with PEX and CUR-loaded MCM-41 MSNs with MASTO (Figure 8A). Similarly, a greater than 60% reduction in cell viability was observed at 8-fold above the FIC concentrations for BR-loaded MCM-41 MSNs with PEX (Figure 8A). Overall, this suggests that the MCM-41-PO_3_^−^ MSN synergistic formulations may provide a less toxic means of translating the observed antimicrobial synergies towards future in vivo and in vitro applications. Of these, the TC- and BR-loaded MCM-41-PO_3_^−^ MSNs with PEX demonstrated synergistic or additive effects towards all the assessed bacterial strains, and thus may provide more broad-spectrum efficacy compared to the QC-loaded MCM-41-PO_3_^−^ MSNs with INDO, which only demonstrated synergistic or additive effects towards MSSA and MRSA.

## 4. Conclusions

Sortase A inhibitors have demonstrated significant promise for the treatment antibiotic resistant microorganisms, with their combination with other antimicrobials offering opportunities for antimicrobial synergy and improved efficacy, in addition to reductions in adverse effects and the development of antimicrobial resistance. However, relatively few SrtAIs have been identified, and of these, many exhibit poor pharmaceutical characteristics and are expensive. The naturally derived SrtAIs studied herein (TC, CUR, QC, and BR) are inexpensive, safe, and readily available, with many used as dietary supplements, but they display poor aqueous solubility. Herein, the use of MCM-41 and a phosphonate-modified analogue, MCM-41-PO_3_^−^, were demonstrated to improve the aqueous solubility of these compounds, affording improved opportunities for their use as SrtAIs. Using 5% (*v*/*v*) DMSO in media as a solvent, these formulations improved their MIC values against medically relevant Gram-positive (*S. aureus*), -negative (*E. coli* and *P. aeruginosa*), and antibiotic-resistant (MRSA) bacteria compared to non-encapsulated formulations, with the phosphonate-modified MSNs generally demonstrating slight improvements in antimicrobial potency and reduced effects on HEK-293 cell viability compared to MCM-41 MSNs. These formulations were also assessed for their capacity to elicit synergistic improvements in antimicrobial efficacy when combined with a library of leading antimicrobial peptides (PEX, MASTO, INDO), with synergistic improvements in antimicrobial efficacy against all tested bacteria observed with TC-loaded MCM-41 when combined with PEX, and similar results, albeit with additive effects on *P. aeruginosa*, observed with the TC-loaded MCM-41-PO_3_^−^ formulation. Furthermore, antimicrobial synergy against MSSA and MRSA was observed for both BR-loaded MSNs when combined with PEX, while synergy against MRSA was observed for both QC-loaded MSNs combined with INDO. These findings suggest that such formulations can significantly improve the antimicrobial potency of both SrtAIs and AMPs, with PEX demonstrating the most synergistic combinations of all the assessed AMPs against the broadest range of bacterial strains. Such improvements will be of significant benefit for the translation of AMPs to the clinic, with the enhanced potency reducing the amount of an AMP required, along with their cost. The combinations also reduced toxicity towards HEK-293 cells, with the MCM-41-PO_3_^−^ formulations demonstrating minimal effects on cell viability. In conclusion, this work demonstrates that MSNs provide a useful strategy for improving the aqueous solubility of the assessed SrtAIs, which provides opportunities for their translation to the market and for the identification of formulations/combinations with beneficial effects on antimicrobial activity and toxicity.

## Data Availability

Data is contained within the article or supplementary material.

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
