# Peer review of "Formulation and Biological Evaluation of Mesoporous Silica Nanoparticles Loaded with Combinations of Sortase A Inhibitors and Antimicrobial Peptides"

_pharmaceutics, 2022, doi:10.3390/pharmaceutics14050986_

Round 1

Reviewer 1 Report

The work reports on MCM41 used for drug incorporation and release. The topic is interesting and it is worth of publication. I suggest revisions as follows.

  • Equations should be numbered in a sequential way.
  • Fig 2. It is provided at very low resolution. Please provide a better figure. Moreover the scale bar should be clearly visible.
  • Fig 3. TGA, the y-axis should be mass or weight % as it starts from 100 %.
  • For DSC the endo or exo direction should be indicated and units in the y-axis have to be included.
  • During the loading procedure, the solvent was evaporated in rotary evaporator. This procedure could be relevant to help the accumulation of drug into the pores (see: Journal of Nanostructure in Chemistry 2021, doi:10.1007/s40097-021-00391-z). This aspect might be discussed. What about the drug release profiles?

Author Response

Reviewer 1

The work reports on MCM41 used for drug incorporation and release. The topic is interesting and it is worth of publication. I suggest revisions as follows.

  • Equations should be numbered in a sequential way.

Response: To address this comment, the equations have been adjusted to be numbered in a sequential way (Equations 1 and 2 on Page 7)

  • Fig 2. It is provided at very low resolution. Please provide a better figure. Moreover the scale bar should be clearly visible.

Response: The resolution of the figure appears to be appropriate in the original files, which were uploaded at submission time. The copy of the manuscript that was sent for review may have been formatted to include a lower resolution copy to reduce the file size. We have changed the color of the scale bars to black or white to ensure maximum contrast against the background for readability and replaced this figure in the manuscript (Page 9) and will upload a new original file with the resubmission.

  • Fig 3. TGA, the y-axis should be mass or weight % as it starts from 100 %.

Response: As per the reviewers comments we have changed the Figure 3 Y-axis caption to Weight (%) (Page 10).

  • For DSC the endo or exo direction should be indicated and units in the y-axis have to be included.

Response: As per the reviewer’s suggestion the endothermic direction has been added to Figure 3 and the units for DSC have been defined as arbitrary units (arb. unit) in the figure and associated caption (Page 10).

  • During the loading procedure, the solvent was evaporated in rotary evaporator. This procedure could be relevant to help the accumulation of drug into the pores (see: Journal of Nanostructure in Chemistry 2021, doi:10.1007/s40097-021-00391-z). This aspect might be discussed. What about the drug release profiles?

Response: We agree that the rotary evaporation method for loading the SrtAIs into the MSNs is relevant with respect to helping to increase the loading of each SrtAI into the MSN pores. This method has been used by other groups for this purpose. We have added a sentence to page 10, section 3.1 describing this ‘In agreement with this study, other studies [19, 33] have demonstrated the capacity of the rotary evaporation loading method to effectively encapsulate both hydrophilic and hydrophobic drugs within the extensive porous network of MSNs’. With respect to drug release profiles, these are beyond the scope of this initial study, which aimed to screen for solubility enhancing effects of MSNs on the poorly aqueous soluble SrtAI library; whether any of these formulations could display synergy with a library of promising antimicrobial peptides; and whether these formulations displayed toxicity towards mammalian cells (see aims Pages 2-3). This extensive and useful data revealed synergistic, safe and effective formulations, which can now be further characterised and optimised in future studies for translation towards clinical use. In relation to such data, we did demonstrate that: the MSN formulations do not exhibit large (> 2 micron) visible to the eye particles, when added to PBS for DLS experiments (Figure 1); the SrtAIs are released from the formulations, with differing effects on improving their solubility over non-formulated SrtAIs (Figure 4); and that these formulations retained or even improved their antimicrobial efficacy (QC and CUR) compared to where the SrtAIs were formulated with organic solvents (5% v/v DMSO) to improve their aqueous solubility (Page 12, Section 3.3).

Reviewer 2 Report

The work from Sitah Alharthi and co-authors concerns the formulation of mesoporous silica nanoparticles containing sortase A inhibitors and their biological evaluation in combination with antimicrobial peptides. The work is well described and the manuscript easy-to-read. The experimental data appear to be correct and the results are well discussed. 

Some comments are:

- line 375: Add a dot at the end of the sentence.

- figure S2: Correct the PO3 to PO3-.

- line 519: Is the value in brackets correct?

Author Response

Reviewer 2

Comments and Suggestions for Authors

  • The work from Sitah Alharthi and co-authors concerns the formulation of mesoporous silica nanoparticles containing sortase A inhibitors and their biological evaluation in combination with antimicrobial peptides. The work is well described and the manuscript easy-to-read. The experimental data appear to be correct and the results are well discussed. 
  • Some comments are:
  • line 375: Add a dot at the end of the sentence.

Response: A full stop was added at the end of the sentence (Page 10).

  • figure S2: Correct the PO3 to PO3-.

Response: This correction has been made to Figure S2 in the supplementary data.

  • line 519: Is the value in brackets correct?

Response: We thank the reviewer for picking up this error. The value that was provided (62.5 micrograms/mL) in the original submission was the MIC value for the TC-loaded phosphonate modified MSNs, rather than the non-modified MSNs (MCM-41). We have made this correction to the manuscript (changed to 125 micrograms/mL), which does not affect the interpretation of the data in this section or in the manuscript as a whole (Page 15, Section 3.5).

Reviewer 3 Report

This manuscript introduces a new strategy of SrtAI-loaded MSN in combination with antimicrobial peptides to improve the antimicrobial potency. A series of SrtAIs have been loaded into MSN, showing better solubility. Some formulations of SrtAI-loaded MSN combined with antimicrobial peptides have shown excellent antimicrobial activity. Some questions need to be addressed before further consideration.

  1. Although MSNs have been widely used as drug carrier for disease therapy, the long-term toxicity in human is still concerned in respect to the significant accumulation of Si elements in major organs, such as liver. Why did authors choose MSN rather than other biocompatible drug carriers, especially liposome.
  2. Both MCM-41 and phosphonate-modified MCM- 82 were chosen in this study, but it is unclear about the necessity. Please explain in “Introduction” section.
  3. The stability and drug release profile of SrtAI-loaded MSN should be further investigated.
  4. It was mentioned “The theoretical MSN loading capacity for each SrtAI was 10% (w/w)(line 365,page 9)”. Please provide more information about this theoretical calculation.
  5. The yield and purity of AMPs are inconsistent in your manuscript (line 422) and supporting information (Figure S3).
  6. It can be seen from Figure 4 that MSN has different effects on the solubility of different hydrophobic drugs. Can you simply explain the reason for this result and summarize what type (such as what functional group) of hydrophobic drugs can effectively improve the solubility when loaded by MSNs.

Author Response

Reviewer 3

  • Although MSNs have been widely used as drug carrier for disease therapy, the long-term toxicity in human is still concerned in respect to the significant accumulation of Si elements in major organs, such as liver. Why did authors choose MSN rather than other biocompatible drug carriers, especially liposome?

Response: Our work showed that MSNs are an attractive strategy for improving the aqueous solubility of the assessed SrtAIs and this platform provides future opportunities for clinical translation. While silica nanoparticles are an emerging nanomaterial, their biocompatibility has been assessed in clinical trials. So far there have been 12 clinical trials/studies where silica nanoparticles have been used for a variety of biomedical applications, including as drug delivery platforms. Based on the data from clinical trials, silica nanoparticles have displayed excellent biocompatibility (Janjua T, et al. Nature Reviews Materials 6.12 (2021): 1072-1074). Furthermore, numerous researchers have studied the long-term safety of silica nanoparticles in animal models and found limited toxicity despite chronic exposure (up to 6 months) even at a high dose (2000 mg/kg) (Int J Nanomedicine. 2014 doi: 10.2147/IJN.S57929, J. Control. Release 2019 doi.org/10.1016/j.jconrel.2019.04.041). We have provided this information in the manuscript on Page 2, where we cite the Nature Reviews Materials publication on their ‘proven biocompatibility in clinical trials’.

Although we appreciate that organic nanoparticles, such as liposomes have enjoyed tremendous success in clinical translation, liposomes can have significant issues with their long-term storage and stability. Working out methods to enable their storage as a dry power and reconstitution, while maintaining the desired shape, and storage stability after reconstitution can be difficult. The silica nanoparticles are more stable with respect to their size in comparison to liposomes.

Thus, we believe that silica nanoparticles have many advantages over liposomes for our particular application. For instance, 1) for nutraceuticals such as the SrtAIs to work effectively, high doses are required, which can be achieved with MSNs; 2) Many lipids degrade rapidly in the GIT and other bodily fluids and phospholipid modifications to improve their stability can lead to costly formulations. As such, we selected to evaluate porous silica nanoparticles for this work.

  • Both MCM-41 and phosphonate-modified MCM- 82 were chosen in this study, but it is unclear about the necessity. Please explain in “Introduction” section.

Response: The reason for using these were: 1) the synthesis and characterisation of both of these MSNs is established in our laboratory, and 2) they both feature an anionic surface charge, allowing us to compare how the additional modification step involved in producing the phosphonate modified MSNs affects the loading capacity of the MSN pores. In addition, if we were to greatly change the surface charge (e.g. by using cationic MSNs), this could provide additional antimicrobial activity and affect the toxicity of these systems, complicating studies to evaluate if formulation improved the potency of the SrtAIs. Thus, at this stage only the anionic charged MSNs were investigated, but investigation of other surface charges would be interesting to study in future.

We have adjusted the introduction (Page 2) to add the following points relating to this question:

‘Previously, our research group and others have shown that different MSN surface functional groups can influence the activity of drugs loaded within the MSN nanopores and the stability of the formulation [17-19]. To date, there has been no extensive comparative investigation of the loading of each member of our SrtAI library into MSNs, and how MSN loading affects their aqueous solubility, antimicrobial activity and toxicity.’

And have added the word anionic into this sentence ‘the loading of anionic MCM-41 and phosphonate-modified MCM-41’ (Page 2) to indicate that we are only investigating MSNs with a negative surface charge in this manuscript.

  • The stability and drug release profile of SrtAI-loaded MSN should be further investigated.

Response: With respect to drug release profiles, these are beyond the scope of this initial study, which aimed to screen for solubility enhancing effects of MSNs on the poorly aqueous soluble SrtAI library; whether any of these formulations could display synergy with a library of promising antimicrobial peptides; and whether these formulations resulted displayed toxicity towards mammalian cells (see aims pages 2-3). This extensive and useful data revealed synergistic, safe, and effective formulations, which can now be further characterised and optimised in future studies for translation towards clinical use. In relation to such data, we did demonstrate that: the MSN formulations do not exhibit large (> 2 micron) visible to the eye particles, when added to PBS for DLS experiments (Figure 1); the SrtAIs are released from the formulations, with differing effects on improving their solubility over non-formulated SrtAIs (Figure 4); and that these formulations retained or even improved their antimicrobial efficacy (QC and CUR) compared to where the SrtAIs were formulated with organic solvents (5% v/v DMSO) to improve their aqueous solubility (Page 12, Section 3.3).

  • It was mentioned “The theoretical MSN loading capacity for each SrtAI was 10% (w/w) (line 365,page 9)”. Please provide more information about this theoretical calculation.

Response: To address this comment, the following statement has been added: ‘The theoretical MSN loading capacity for each SrtAI was 10% (w/w), based on the rotary evaporation formulation method, where 10 mg of each individual SrtAI was added to 90 mg of each MSN’ (Page 9).

  • The yield and purity of AMPs are inconsistent in your manuscript (line 422) and supporting information (Figure S3).

Response: An error with the method used to calculate the yield and purity values was identified during manuscript preparation. This was fixed in the manuscript prior to submission, but unfortunately was not noted in the supporting information prior to submission. We thank the reviewer for noting this error, which has now been fixed in the supporting information to provide the same information as the revised manuscript (Figure S3).

  • It can be seen from Figure 4 that MSN has different effects on the solubility of different hydrophobic drugs. Can you simply explain the reason for this result and summarize what type (such as what functional group) of hydrophobic drugs can effectively improve the solubility when loaded by MSNs.

Response: Thanks to the reviewer for this comment. In this study we only looked at MCM-41 and its phosphate-modification in this paper as the aims were more about improving the solubility of the SrtA inhibitors and investigating if these formulations could demonstrate synergies with a library of different antimicrobial peptides than extensive assessment of different surface-modifications. As we were able to achieve the aims of this study, studying the effects of different charged functional groups may be a useful future study to see if altering the surface charge can improve the ability to associate the antimicrobial peptides with the delivery systems, and can still improve the delivery characteristics of the SrtAIs as well as provide synergy. We note in Figure 4 that there was not a large difference between the MCM-41 and phosphonate-modified MCM-41 with respect to their ability to improve the aqueous solubility of each SrtAI. Both of these types of MSN feature anionic surface charges (with the phosphonate-modified version having a slightly more negative Zeta-potential) (Table 1), and thus may demonstrate the ability to better bind to drugs that feature a cationic charge. Of the molecules that we studied, only berberine features a cationic charge, and in its case, we saw similar loading to the other compounds – particularly with the MCM-41 MSNs, where loading capacities between 7.6 and 9.8 % were observed compared to 8.7% for berberine. Furthermore, some of these molecules are uncharged (e.g. trans-chalcone) or feature phenolic protons, which may be partly deprotonated at neutral pH, resulting in anionic charged molecule that may be repelled by the anionic charge associated with the MSNs used herein. Despite this, each of these compounds demonstrated similar loading, and the solubility enhancing effect of the MSNs could not be definitively related back to the functional groups that were present in each molecule or MSN with any reliability. Hence this has not been extensively discussed.

Reviewer 4 Report

Comments: The work is an interesting manuscript,  However there are few conceptual issues in this article, Would you please clarify following arguments?

  • MCM-41 NPs were  modified  by phosphonate groups instead of amine groups ((3-aminopropyl)- triethoxysilane . This provide unclear  state if the purpose to use them for  loading  the negative charge materials such as CUR, QR., Authors should to provide clear explanation why used  phosphonate modification instead of amine modification
  • FTIR spectra for all used materials, should to be applied to understand clearly chemical interaction between MCM-41 NPs   and SrtAI(TC, CUR, QC, and BR)
  • Scheme to illustrate fabrication , modification and loading capacity of MCM-41 NPs, should to be provided.
  • In line 411, it is written “The enhanced solubility profile for each SrtAI-loaded MSN formulation is likely explained by the conversion of the compounds from a crystalline, to an amorphous state upon MSN encapsulation”. Authors did not provide clear experiment to confirm such this result. XRD is recommended.
  • SrtAI-loaded MCM-41 obtained mostly good negative charge (zeta potential) . authors should to explain how such these materials can adhere and internalize to bacteria used in this study. TEM should to be used to explain cellular internalization
  • Authors provide cytotoxicity assay by using just one cell line (HEK-293). Two different cell lines are recommended.
  • Authors did not provide clear description for what is the novelty of the current work, if it is already published previously in the same approach.

Author Response

Reviewer 4

Comments: The work is an interesting manuscript,  However there are few conceptual issues in this article, Would you please clarify following arguments?

  • MCM-41 NPs were  modified  by phosphonate groups instead of amine groups ((3-aminopropyl)- triethoxysilane . This provide unclear  state if the purpose to use them for  loading  the negative charge materials such as CUR, QR., Authors should to provide clear explanation why used  phosphonate modification instead of amine modification

Response: In this study we didn’t focus on extensive investigations of different surface functionalisation chemistries and only looked at the MCM-41 and phosphate-modification in this manuscript as the aims were more about improving the solubility of the SrtA inhibitors and investigating if these formulations could demonstrate synergies with a library of different antimicrobial peptides. As we achieved these aims, studying the effects of different charged functional groups may be a useful future study to see if these improve the ability to associate the antimicrobial peptides with the delivery systems, and can still improve the delivery characteristics of the SrtAIs as well as provide synergy.

In addition, the phosphonate-modification was studied herein as it retains the anionic surface charge of MCM-41, allowing us to compare how the additional modification step involved in producing the phosphonate modified MSNs affects the loading capacity of the MSN pores. In addition, if we were to greatly change the surface charge (e.g. by using cationic MSNs), this could provide additional antimicrobial activity and affect the toxicity of these systems, complicating studies to evaluate if formulation improved the potency of the SrtAIs. Thus, at this stage only the anionic charged MSNs were investigated, but investigation of other surface charges would be interesting to study in future.

We have adjusted the introduction (Page 2) to add the following points relating to this question:

‘Previously, our research group and others have shown that different MSN surface functional groups can influence the activity of drugs loaded within the MSN nanopores and the stability of the formulation [17-19]. To date, there has been no extensive comparative investigation of the loading of each member of our SrtAI library into MSNs, and how MSN loading affects their aqueous solubility, antimicrobial activity and toxicity.’

And have added the word anionic into this sentence ‘the loading of anionic MCM-41 and phosphonate-modified MCM-41’ (Page 2) to indicate that we are only investigating MSNs with a negative surface charge in this manuscript.

  • FTIR spectra for all used materials, should to be applied to understand clearly chemical interaction between MCM-41 NPs   and SrtAI (TC, CUR, QC, and BR)

Response: FT-IR spectra were obtained for each SrtAI, MSN and SrtAI-loaded MSN. This data has been added as supporting information (Figure S21) for the reviewer and readers’ reference. To incorporate this into the manuscript, we have added the sentence ‘A Perkin Elmer Spectrum TWO (Liantrisant, UK) Fourier-transform infrared spectrometer (FT-IR) was used to investigate interactions between MSNs and SrtAIs’ in section 2.2 (Page 4) and ‘FT-IR spectra were obtained for 2mg of each sample (SrtAIs, MSNs, and SrtAI-loaded MSNs) over a 4000 to 400 cm-1 wavenumber range to investigate interactions between MSNs and SrtAIs (data provided in Figure S21)’. The DLS (Figure 1), TEM (Figure 2), TGA/DSC (Figure 3), SrtAI solubility assessment (Figure 4),  and N2-adsorption-desorption data (Table 2), which are discussed in the manuscript, provide good data to demonstrate the loading of the MSNs with the SrtAIs.

  • Scheme to illustrate fabrication , modification and loading capacity of MCM-41 NPs, should to be provided.

  Response: A scheme indicating the fabrication, surface-modification, and loading of the MSNs has been added to the manuscript (Scheme 1, Page 4).

  • In line 411, it is written “The enhanced solubility profile for each SrtAI-loaded MSN formulation is likely explained by the conversion of the compounds from a crystalline, to an amorphous state upon MSN encapsulation”. Authors did not provide clear experiment to confirm such this result. XRD is recommended.

Response: The rotary evaporation procedure for loading MSNs with SrtAIs helps to prevent crystallisation of the SrtAIs. This is because the SrtAIs are initially dissolved in an organic solvent (and thus is not in a crystalline form), which as the volume is reduced, helps to concentrate the SrtAI molecules into the MSN pores. As the solvent evaporates, the drug then precipitates within the nanopores, which slows crystallisation, leading to an amorphous solid dispersion of drug within the MSN pores. This is evident in Figure 3 where sharp endothermic DSC peaks are seen for trans-chalcone (61 C), curcumin (177 C), quercetin (317 C) and berberine (190 C) alone. Similar (albeit smaller) peaks can be seen with physical mixtures of MSNs with SrtAIs (except quercetin), while these peaks disappear, or at least are greatly reduced) after the drug is loaded into the MSN pores. This data is consistent with the literature for loading curcumin into MSNs and conversion to an amorphous form (Jambhrunkar S, et al. RSC Adv 2014;117:520 – 91 citations), which included XRD data. As well as other papers, which have used DSC data for this purpose (e.g. Biswas N, et al. Eur J Pharm Sci 2017;99;152). The enhanced solubility observed in Figure 4 also provides evidence for this. As there is no additional XRD data in this paper, the wording of this section specifies that this enhanced solubility is ‘likely’ explained by conversion of the compounds from crystalline to amorphous, rather than providing a definitive statement.

  • SrtAI-loaded MCM-41 obtained mostly good negative charge (zeta potential) . authors should to explain how such these materials can adhere and internalize to bacteria used in this study. TEM should to be used to explain cellular internalization

Response: As sortase A is associated with the bacterial cell surface, and thus is exposed to the external environment, the SrtAIs do not require internalisation to exert their activity (discussed in the introduction, on Page 2). In addition, for these compounds to be able to fit into the active site, and inhibit SrtA, they need to be released in a soluble form from the silica, so we are using the silica to improve the aqueous solubility of the SrtAIs, and not to improve their internalisation into bacteria. Thus, there is no need to use TEM to explain cellular internalisation for this application, or to discuss this further. In addition, we have demonstrated that each SrtAI can be released from these formulations to permit their interaction with sortase A in Figure 4, and that each of these formulations retains (Figure 5), or even enhances the antimicrobial efficacy (CUR and QC-loaded MSNs discussed in section 3.3) of the SrtAIs when compared to 5% v/v DMSO-formulated SrtAIs.

  • Authors provide cytotoxicity assay by using just one cell line (HEK-293). Two different cell lines are recommended.

Response: The cell viability assay was a preliminary toxicity screen. HEK293 cells were used as they are commonly used to screen for toxicity of compounds against mammalian cells. This assay provides a simple means to compare the toxicity of each tested component individually, and when delivered as a mixture. For example, we might identify that the combination of AMPs with MSN SrtAI formulations could not only be synergistic for their antimicrobial activity, but also in terms of toxicity. For progressing the leading formulations towards clinical studies, more detailed assessment against multiple cell lines, of different lineages, and the use of different toxicity screens will be performed, but this is beyond the scope of these initial studies

  • Authors did not provide clear description for what is the novelty of the current work if it is already published previously in the same approach.

Response: The work presented in this paper has not been previously published. Firstly, we have investigated the ability to formulate a library of four different SrtAIs in MSNs, using two different MCM-41 type mesoporous nanoparticles. Not all of these compounds have been assessed for formulation with MCM-41 type MSNs, and especially with the phosphonate-modified analogs. In addition, we are the first group comprehensively compare how these MSN formulations affect the antimicrobial activity of this library of SrtAIs against a variety of disease relevant (E. coli, S. aureus, and P. aeruginosa) and drug-resistant (methicillin-resistant S. aureus) bacterial strains. Further, we have provided the first comprehensive data demonstrating the potential for synergy between leading antimicrobial peptides and these formulations. This has not been published previously, and represents an important step in progressing these important new sources of antibiotic molecules/formulations towards clinical use by reducing the amount of these species required for their activity (i.e. synergy improves their potency), improving their spectrum of activity, and offers a means to reduce their toxicity. Further, as both agents have a low risk of resistance developing, these combinations represent an important approach for overcoming multi-drug resistant bacterial infections.

We have discussed many of these concepts within the manuscript, in particular at the end of the introduction:

‘This is of significance as AMPs have a novel mechanism of action (in comparison to current antibiotics), which omits the development of resistance, but due to their peptide nature, exhibit significant costs. Thus, improving their potency would be of significance, as this would reduce the dose of an AMP required for efficacy, reducing their cost and toxicity, and by encapsulating the AMP, would help protect them from degradation. Thus, these characteristics would improve the commercial viability of AMPs as alternative antimicrobial treatments for superbug infections.’

In addition, the following sentence has been added to the introduction (Page 2):

‘Previously, our research group and others have shown that different MSN surface functional groups can influence the activity of drugs loaded within the MSN nanopores and the stability of the formulation [17-19]. To date, there has been no extensive comparative investigation of the loading of each member of our SrtAI library into MSNs, and how MSN loading affects their aqueous solubility, antimicrobial activity, and toxicity.’

Round 2

Reviewer 1 Report

revised version can be published

Reviewer 3 Report

Authors have addressed all the questions.

Reviewer 4 Report

Manuscript was revised point by point according to reviewer comments.

Manuscript is more acceptable NOW.